

# Comprehensive screening of low nitrogen tolerant maize based on multiple traits at the seedling stage

Jianjia Miao, Fei Shi, Wei Li, Ming Zhong, Cong Li and Shuisen Chen

College of Bioscience and Biotechnology, Shenyang Agricultural University, Shenyang, Liaoning, China

## ABSTRACT

**Background**. Plants tolerant to low nitrogen are a quantitative trait affected by many factors, and the different parameters were used for stress-tolerant plant screening in different investigations. But there is no agreement on the use of these indicators. Therefore, a method that can integrate different parameters to evaluate stress tolerance is urgently needed.

**Methods**. Six maize genotypes were subject to low nitrogen stress for twenty days. Then seventeen traits of the six maize genotypes related to nitrogen were investigated. Nitrogen tolerance coefficient (NTC) was calculated as low nitrogen traits to high nitrogen traits. Then principal component analysis was conducted based on the NTC. Based on fuzzy mathematics theory, a D value (decimal comprehensive evaluation value) was introduced to evaluate maize tolerant to low nitrogen.

**Results**. Three maize (SY998, GEMS42-I and GEMS42-II) with the higher D value have better growth and higher nitrogen accumulation under low nitrogen conditions. In contrast, Ji846 with the lowest D value has the lowest nitrogen accumulation and biomass in response to nitrogen limitation. These results indicated that the D value could help to screen low nitrogen tolerant maize, given that the D value was positively correlated with low nitrogen tolerance in maize seedlings.

**Conclusions**. The present study introduced the D value to evaluate stress tolerance. The higher the D value, the greater tolerance of maize to low nitrogen stress. This method may reduce the complexity of the investigated traits and enhance the accuracy of stress-tolerant evaluation. In addition, this method not only can screen potentially tolerant germplasm for low-nitrogen tolerance quickly, but also can comprise the correlated traits as many as possible to avoid the one-sidedness of a single parameter.

## INTRODUCTION

Nitrogen is an essential nutrient for plant growth and development, and it is also a major driving force for crop productivity improvement. Screening and developing varieties with nitrogen efficient crop plays a pivotal role in agriculture's sustainable development (*Liu et al., 2022*). Nitrogen uptake and utilization efficiency for grain production depend on those processes associated with absorption, translocation, assimilation and redistribution

Corresponding author
Shuisen Chen,
shuisenchen@syau.edu.cn

of nitrogen to operate effectively (*Masclaux-Daubresse et al., 2010*; *Xu, Fan & Miller, 2012*). Plants uptake the nitrate through the low- and high-affinity nitrate transporters (*Fan et al., 2017*; *Vidal et al., 2020*). While the ammonium uptake was mediated by the saturable high-affinity (ammonium transporters) and the nonsaturable low-affinity (aquaporins or cation channels) uptake system (*Tegeder & Masclaux-Daubresse, 2018*). The nitrate reductase (NR), glutamine synthetase (GS) and glutamate synthase (GOGAT) were the key enzymes for nitrogen assimilation that indirectly affect the metabolism, allocation and remobilization of nitrogen in plants (*Lea et al., 2006*; *Martin et al., 2006*).

Maize (*Zea mays* L.) is an important food and forage crop in the world, as well as an important energy crop (*Yin et al., 2014*). Moreover, maize is the crop with the highest production among all crops and is also the crop with the greatest demand for nitrogen (*Sivasankar et al., 2012*). Due to the differences in nitrogen absorption and utilization among maize genotypes (*Harvey, 1939*), more focus was paid to screening and improving nitrogen efficiency (*Hirel et al., 2007*). The greater differences in growth and yield among the maize lines and hybrids were associated with both the nitrogen uptake and utilization efficiency in response to low nitrogen stress (*Hirel & Gallais, 2011*). The root architecture of maize is a key factor affecting the nitrogen absorption, and more photosynthate will distribute to the root to enhance the root surface of the nitrogen-efficient maize under nitrogen limitation (*Sinclair & Vadez, 2002*), The absorption of nitrogen in roots requires the involvement of the high-affinity nitrogen transporter (NRT2 and AMT1), especially under the nitrogen limitation (*Dechorgnat et al., 2019*). Among the four *ZmNRT2* identified in the maize genome, only *ZmNRT2;1* and *ZmNRT2;2* have proven to be correlated with nitrate ($NO_3^-$) uptake capacity (*Plett et al., 2010*; *Garnett et al., 2013*). Furthermore, *ZmAMT1;1a* and *ZmAMT1;3* have been identified to encode functional ammonium transporters for high-affinity ammonium uptake in maize roots (*Gu et al., 2013*).

Nitrogen has significantly influenced the productivity and characteristics of maize (*Teixeira et al., 2014*). However, the higher nitrogen fertilizer application led to negative effects on the ecological environment because of lower nitrogen uptake and utilization efficiencies of plants. Hence, it is increasingly important to screen nitrogen stress-tolerant plants or explore nitrogen-efficient plants that are more efficient at nitrogen utilization and better suited to nitrogen limitation. Plants tolerant to low nitrogen is a quantitative trait affected by many factors which result in high cost both in time and resources of measuring certain traits for screening nitrogen-tolerant maize. Fortunately, principal component analysis is a quantitatively rigorous method for multivariate datasets simplification. It can transform more original indicators into several new relatively independent comprehensive indicators. The absolute subordination of elements to sets was broken in the theory of fuzzy mathematics. Subordinate function analysis was one of effective ways used in comprehensive evaluation of abiotic stresses (*Shi et al., 2010*). To comprehensively evaluate the low nitrogen tolerance of maize varieties more conveniently and effectively, a D value was introduced based on the fuzzy mathematics theory. Our study would provide a comprehensive and dependable method for evaluating low tolerance in maize.

## MATERIAL AND METHODS

### Plant material, growth and treatment conditions

The six maize, GEMS42-I, Ji846, SY998, CML223, CML114 and GEMS42-II, with a significant difference in grain yield and nitrogen tolerance were used in the present study. The surface-sterilized seeds germinated on wet sand in the culture room. Then, the 4-day-old seedlings were transferred into the nutrient solution for continuing growth. The complete basal nutrient solution contained 0.24 g/L $NH_4NO_3$, 0.50 g/L $MgSO_4$, 0.15 g/L KCl, 0.36 g/L $CaCl_2$, 0.05 mM EDTA-Fe and a microelement solution (*Hoagland & Arnon, 1950*). The nutrient solution containing 1/10 N of the complete nutrient solution was used for low nitrogen treatment (-N), and the seedlings growing under the complete nutrient were used as control (+N). Keep the culture room parameter as follow: 16 h of light (300–320 $\mu$mol $m^{-2}$ $s^{-1}$) at 24 °C and 8 h of darkness at 22 °C photoperiods, and relative humidity of 65–80%. Roots and leaves of all six maize were harvested separately after growing under low nitrogen conditions for 20 days. Each treatment was replicated three times.

### Biomass and phenotypic characteristics of the root system

Root was floated in the water and scanned using the scanner (Epson Expression 11000XL) to get the image. The root total length, root volume, root surface area and root average diameter were calculated with Tennant's statistical method in WinRHIZO Pro software (Version 2.0, 2005; Regent Instrument Inc., Quebec, Canada) as previous study (*Altaf et al., 2022*). The seedlings were washed with distilled water, and the fresh weight (FW) was measured after drying with bibulous paper.

### Measurement the $NO_3^-$ and $NH_4^+$ content of the seedlings

For nitrates ($NO_3^-$) determination, roots and shoots (approximately 0.5 g FW) were cut into pieces and suspended in 5 mL boiling water for 10 min (*Tang et al., 2013*). Then the supernatant was diluted to 25 mL. The assay mixture containing 0.1 mL samples and 0.4 mL 5% salicylic acid-sulfuric acid, was incubated at 20 °C for 20 min, then mixed with 9.5 mL 8% NaOH (w/v). Its absorbance was measured at 410 nm wavelength.

The ammonium ($NH_4^+$) of root and shoot were extracted by homogenizing in 0.3 mM $H_2SO_4$ (pH 3.5). After centrifugation at 3,900 g for 10 min, the supernatant was collected using for the determination of ammonium ($NH_4^+$) content as previously described (*Lin & Kao, 1996*). After $NO_3^-$ the and $NH_4^+$ determination, the root nitrogen accumulation, shoot nitrogen accumulation and total plant nitrogen accumulation were calculated.

### Enzyme activity assays

Approximately 0.5 g of fresh roots were homogenized with 10 mM Tris–HCl buffer (pH 7.6) containing 1 mM $MgCl_2$, 1 mM EDTA and 1 mM $\beta$-mercaptoethanol in a chilled pestle and mortar. After centrifugation at 15,000 g for 30 min (4 °C), the supernatant was used as an enzyme extract (*Ren et al., 2017*).

The whole extraction procedure was carried out at 4 ° C.

For GS (EC6.3.1.2) activity assayed, a 1.0 mL reaction mixture (pH 8.0) contained 80 $\mu$mol Tris–HCl buffer, 40 $\mu$mol L-glutamic acid, 8.0 $\mu$mol ATP, 24 $\mu$mol $MgSO_4$,

and 16 µmol $NH_2OH$ and enzyme extract. The enzyme extract was added to initiate the reaction. After incubation for 30 min at 30 °C, the reaction was stopped by adding two mL 2.5% (w/v) $FeCl_3$ and 5% (w/v) trichloroacetic acid in 1.5 M HCl. After centrifugation at 3,000 g for 10 min, the absorbance of the supernatant was measured at 540 nm. GS activity was expressed as 1.0 µM L-glutamate $\gamma$-monohydroxamate (GHA) formed $g^{-1}$ FW $h^{-1}$, with µmol GHA $g^{-1}$ FW $h^{-1}$.

For GOGAT (EC1.4.7.1) activity assayed, a three mL reaction solution was prepared with 25 mM Tris–HCl buffer (pH 7.6), which contained 0.5 mL enzyme extract, 0.05 mL 0.1 M 2-oxoglutarate, 0.1 mL 10 mM KCl, 0.2 mL 3 mM NADH and 0.4 mL 20 mM L-glutamine. The reaction was initiated by adding L-glutamine immediately following the enzyme preparation. The decrease in absorbance was recorded for 3 min at 340 nm. The GOGAT activity was expressed as µmol NADH $g^{-1}$ FW $h^{-1}$.

NR (EC1.7.1.1) activity was determined according to Wojciechowska et al. with minor modifications (*Wojciechowska et al., 2016*). The NR activity was expressed as µg $NO_2^-$ $g^{-1}$ FW $h^{-1}$.

## Quantitative RT-PCR analysis

Total RNA was isolated using TRIzol reagent (Invitrogen, CA, USA) and then first-strand cDNA was synthesized using the M-MLV Reverse Transcriptase (Promega, WI, USA) according to the manufacturer's instructions. For the quantitative real-time PCR (qRT-PCR) experiment, 20 µL reaction components were prepared according to the manufacturer's protocol for SYBR Green Real Master Mix (TIANGEN, Beijing, China). Using *GAPDH* (glyceraldehyde-3-phosphate dehydrogenase) as the endogenous control. Real-time PCR was conducted on the CFX96$^{TM}$ Real-Time PCR Detection System (Bio-Rad, CA, USA), and the primer pairs used for quantitative RT-PCR were shown in supplemental Table S1.

## Data analysis and D value calculation

The standard deviation (SD) was used to express the sample variability in the present study. All analyses of significance were conducted at the $p < 0.05$ level. Considering that the biological differences among the different maize genotypes, evaluation of the low nitrogen tolerance of maize by NTC may be more reasonable. The NTC was calculated as $NTC = \frac{\text{low nitrogen trait}}{\text{high nitrogen trait}}$. Then principal component analysis was conducted based on the NTC in SPSS (Statistical Product and Service Solutions) software (version 18.0). The principal component (PCi, i = 1, 2 …n) with eigenvalue ($\lambda_i$, i = 1, 2 …$n$) >1 was selected as new index. PC$i$ is the $i$-the principal component. $\lambda_i$ is the eigenvalue of the $i$-the principal component. The eigenvalue ($\lambda_i$, i = 1, 2 …n) and factor score (FACi, i = 1, 2 …$n$) were present in the results of the principal component analysis. The principal component value $Xi$ was calculated as $Xi = FAC\, i \times \sqrt[2]{\lambda_i}$ ($i$ = 1, 2 …$n$). Then the subordinate function value was calculated as $U(Xi) = \frac{Xi - Xmin}{Xmax - Xmin}$ ($i$ = 1, 2 …$n$). $Xmax$ and $Xmin$ represent the maximum and minimum value of the $i$th principal component, respectively. The weight coefficient was calculated as $W(i) = \frac{Pi}{\sum_{i=1}^{n} Pi}$ ($i$ = 1, 2 …$n$). $Pi$ represents the proportion of variance explained by the $i$-the principal component. Finally, the D value was calculated as $D = \sum_{i=1}^{n} [U(Xi) \times W(i)]$ ($i$ = 1, 2 …$n$).

## Statistical analysis

The standard deviation was used to express the sample variability in the present study. The significance of differences was conducted using SAS 9.2 (SAS Institute, Cary, NC, USA). Data were subjected to ANOVA using PROC LSD ($p < 0.05$) in SAS.

# RESULTS

## Plant physiological changes in response to nitrogen stress

Low nitrogen (ca. 0.3 mM $NH_4NO_3$) significantly inhibited the growth of Ji846 but not of the other five genotypes of maize (Fig. 1), and even significantly increased the root biomass of SY998 and GEMS42-II by 46% and 66%, respectively (Fig. 1C). Consider that root is the primary organ for water and nutrients capturing, the morphology of root is investigated by WinRHIZO Pro software. All of the root total length, root surface and root volume of Ji846 were significantly decreased in response to low nitrogen stress, which decreased by 56%, 65% and 74%, respectively (Fig. 2). Root diameter of the six maize were decreased in response to low nitrogen stress (Fig. 2B). In contrast, the root total length, root surface and root volume of SY998 and GEMS42-II were prominently increased under low nitrogen (Figs. 2B, 2D, 2E). Low nitrogen not affected the root total length, root surface and root volume of GEMS42-I, CML223 and CML114 (Figs. 2B, 2D, 2E). These results indicated that Ji846 was sensitive to low nitrogen stress.

## $NO_3^-$ and $NH_4^+$ content in maize

Nitrate ion ($NO_3^-$) and ammonium ion ($NH_4^+$) are the main form of nitrogen for plant absorption. Maize can uptake both nitrate and ammonium. Both the $NO_3^-$ and $NH_4^+$ contents were significantly decreased in all maize genotypes under low nitrogen stress (Fig. 3). However, only the root $NO_3^-$ content of SY998 was significantly higher than the other genotype under low nitrogen (Fig. 3A). Under low nitrogen stress, Ji846 has the lowest $NH_4^+$ content both in root and shoot, while the GEMS42-I and SY998 have the highest $NH_4^+$ content both in root and shoot (Figs. 3C and 3D). The lowest nitrogen accumulation was observed in Ji846 (8.78 $\mu$g N plant$^{-1}$) under nitrogen limitation, no matter in the root, shoot, or total plant (Table 1). While the highest nitrogen accumulation was observed in SY998 (153.87 $\mu$g N plant$^{-1}$) under nitrogen limitation (Table 1). Interestingly, the $NO_3^-$ content was higher than the $NH_4^+$ content of all six maize, irrespective of the nitrogen nutritional status of the plants. In the high nitrogen condition, the $NO_3^-$ of root and shoot was 12.3 and 10.4 times the $NH_4^+$ in Ji846, respectively. While under the low nitrogen condition, the $NO_3^-$ of root and shoot was 7.1 and 6.6 times the $NH_4^+$ in Ji846, respectively (Fig. 3).

## Expression of the nitrate and ammonium transporter genes

The expression of *ZmNRT2;1* and *ZmNRT2;2* in Ji846 and SY998 were significantly increased under low nitrogen conditions (Figs. 4A and 4B). The expression of *ZmNRT2;1* under low nitrogen was nine and three times the high nitrogen condition in Ji846 and SY998, respectively. The expression of *ZmNRT2;2* under low nitrogen was 41 and 11 times the high nitrogen condition in Ji846 and SY998, respectively (Figs. 4A and 4B). In addition,

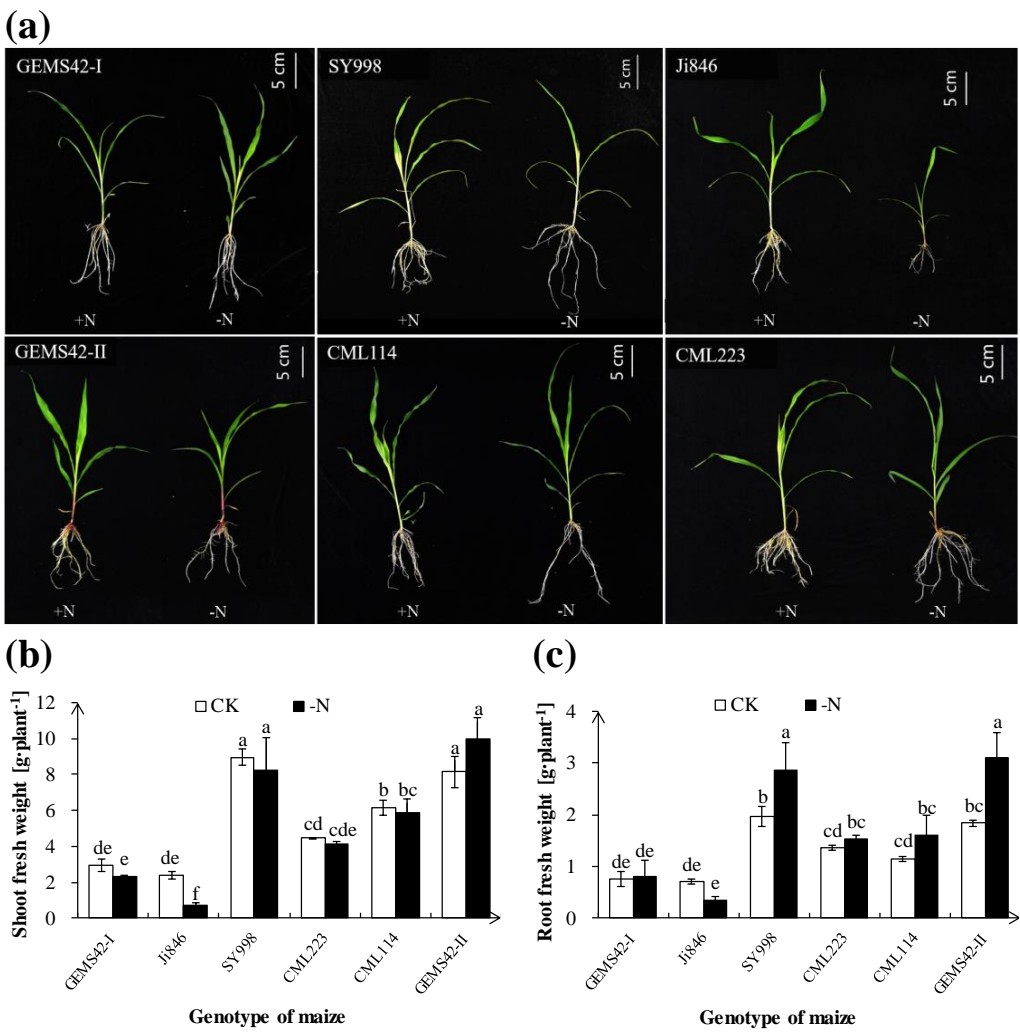

**Figure 1** **The morphological appearance (A), roots biomass (B) and shoots biomass (C) of the different genotype maize in response to low nitrogen stress.** The different genotype maize subjected to nitrogen stress for three weeks. The +N and –N represent the seedling under high nitrogen (3 mM $NH_4NO_3$) and low nitrogen (0.3 mM $NH_4NO_3$), respectively. Values represent the mean ± SD, bars with different letters show significant differences (ANOVA, LSD, $P < 0.05$).

the expression of *ZmNRT2;2* in CML223 was also significantly increased (5 times) under nitrogen limitation (Fig. 4B). For the ammonium transporters genes, the expression of *ZmAMT1;1a* was significantly increased under nitrogen limitation in GEMS42-I, Ji846, SY998 and GEMS42-II, which increased 16, 10, 20 and 3 times, respectively (Fig. 4C). The expression of *ZmAMT1;3* was significantly increased under nitrogen limitation in Ji846, CML223, CML114 and GEMS42-II, which varied from 1.4 to 4.3 times (Fig. 4).

## Nitrogen metabolism-related key enzymes activity assay

Low nitrogen significantly decreased the activities of key nitrogen metabolism enzymes in some maize. The greatest reduction in the activities of NR (approximately 82%) in Ji846

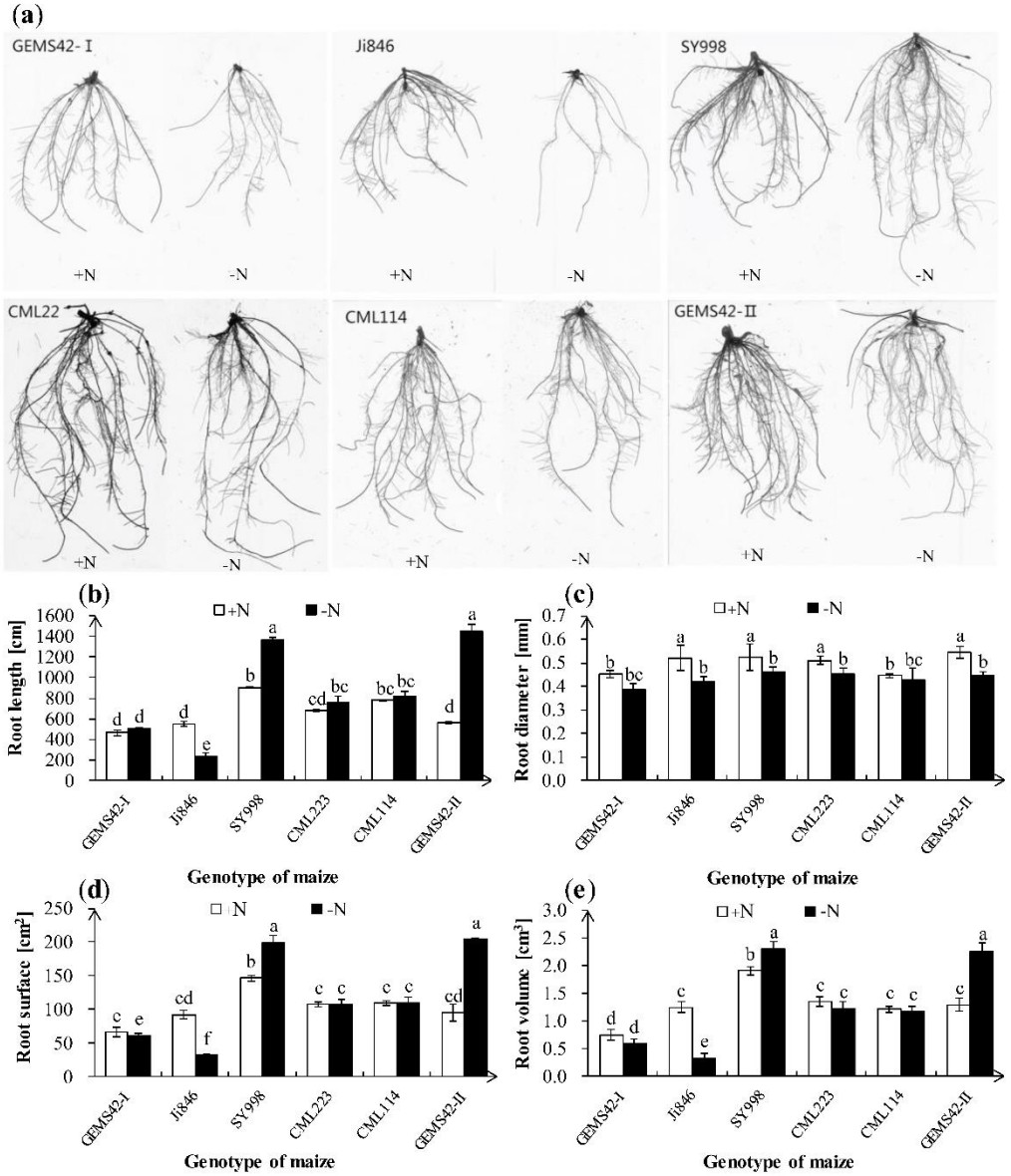

**Figure 2 Effects of low nitrogen stress on root morphology.** (A) Morphological appearance of roots, (B) root total length, (C) root average diameter, (D) root surface area and (E) root volume. The +N and –N represent the seedling under high nitrogen (3 mM $NH_4NO_3$) and low nitrogen (0.3 mM $NH_4NO_3$), respectively. Values represent the mean ± SD of ten seedlings in each treatment. Bars with different letters show significant differences at $p < 0.05$.

(Fig. 5A), GS (approximately 88%) in Ji846 (Fig. 5B), (GOGAT approximately 56%) in CML223 (Fig. 5C). The NR activities of GEMS42-I, CML223 and CML114 were decreased 60.3%, 68.6% and 48.6%, respectively (Fig. 5A). In addition, the lower activities of NR and GS were observed in SY998, CML223 and GEMS42-II, irrespective of the nitrogen condition (Figs. 5A and 5B). Nitrogen limitation affected the activity of GOGAT less than

Peer

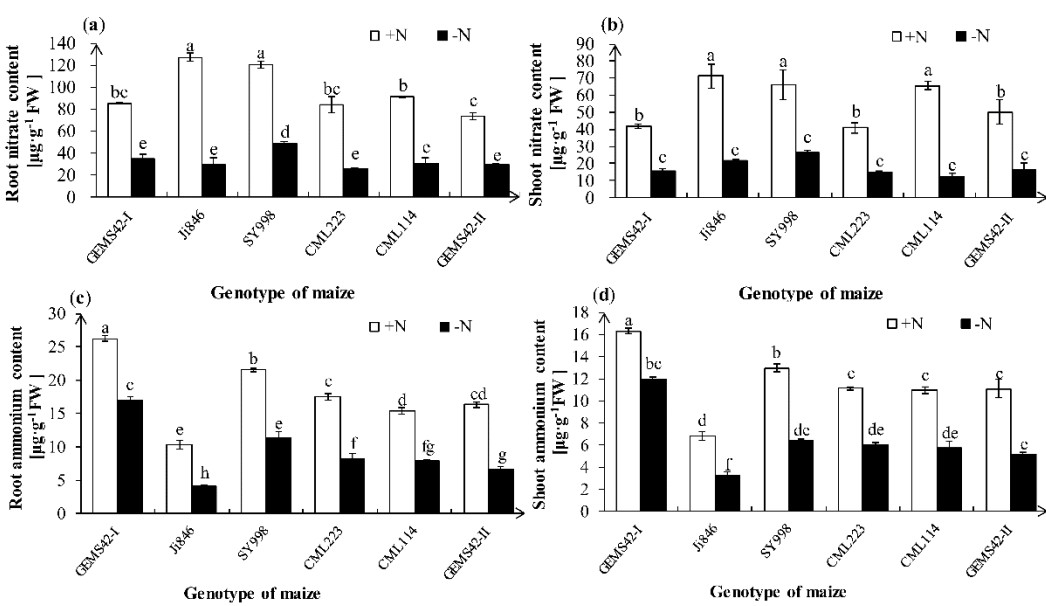

**Figure 3** **Effects of low nitrogen stress on nitrate ($NO_3^-$) and ammonium ($NH_4^+$) content of maize.**
Fig. 3 Effects of low nitrogen stress on nitrate ($NO_3^-$) and ammonium ($NH_4^+$) content of maize. (A) Root nitrate ($NO_3^-$) content, (B) shoot nitrate ($NO_3^-$) content, (C) root ammonium ($NH_4^+$) content, (D) shoot ammonium ($NH_4^+$) content. The +N and –N represent the seedling under high nitrogen (3 mM $NH_4NO_3$) and low nitrogen (0.3 mM $NH_4NO_3$), respectively. Values represent the mean $\pm$ SD of ten seedlings in each treatment. Bars with different letters show significant differences at $p < 0.05$.

**Table 1** **The nitrogen accumulation of each maize.**

| Maize Genotype | Root ($\mu g$ N plant$^{-1}$) | | Shoot ($\mu g$ N plant$^{-1}$)] | | Total plant ($\mu g$ N plant$^{-1}$) | |
| --- | --- | --- | --- | --- | --- | --- |
| | CK | -N | CK | -N | CK | -N |
| GEMS42-I | $29.74 \pm 0.77$e | $16.85 \pm 0.75$ g | $65.22 \pm 0.18$de | $30.15 \pm 1.39$f | $94.96 \pm 0.95$f | $47.00 \pm 0.64$ g |
| Ji846 | $34.96 \pm 0.24$d | $3.25 \pm 0.70$ h | $68.68 \pm 6.60$d | $5.52 \pm 0.01$ g | $103.64 \pm 6.35$ef | $8.78 \pm 0.70$ h |
| SY998 | $86.72 \pm 1.04$a | $56.33 \pm 0.46$b | $206.48 \pm 23.85$a | $97.54 \pm 1.68$c | $293.20 \pm 22.80$a | $153.87 \pm 1.22$d |
| CML223 | $44.07 \pm 3.05$c | $18.47 \pm 0.28$fg | $79.82 \pm 0.93$cd | $33.28 \pm 0.06$f | $123.89 \pm 2.12$e | $51.75 \pm 0.34$ g |
| CML114 | $37.45 \pm 0.22$d | $20.91 \pm 2.76$f | $143.35 \pm 0.21$b | $42.96 \pm 3.36$ef | $180.80 \pm 0.01$c | $63.87 \pm 0.61$ g |
| GEMS42-II | $42.40 \pm 0.53$c | $36.39 \pm 1.79$d | $185.37 \pm 24.67$a | $76.15 \pm 15.44$cd | $227.77 \pm 24.14$b | $112.55 \pm 17.24$ef |

that of NR and GS. The GOGAT activities only decreased in CML223 and GEMS42-II in response to nitrogen limitation (Fig. 5C).

## Principal component analysis based on the nitrogen tolerance coefficient (NTC)

The first four principal components jointly explain the major part of the total variance (96.8%), being PC1 responsible for 47.4%, PC2 for 21.6%, PC3 for 15.5% and PC4 for 12.3% of the total variance, respectively (Table 2). The eigenvalue of PC1, PC2, PC3 and PC4 were 8.054, 3.676, 2.627 and 2.091, respectively (Table 2). The factor score of the four principal components of each maize was directly extracted from the principal component

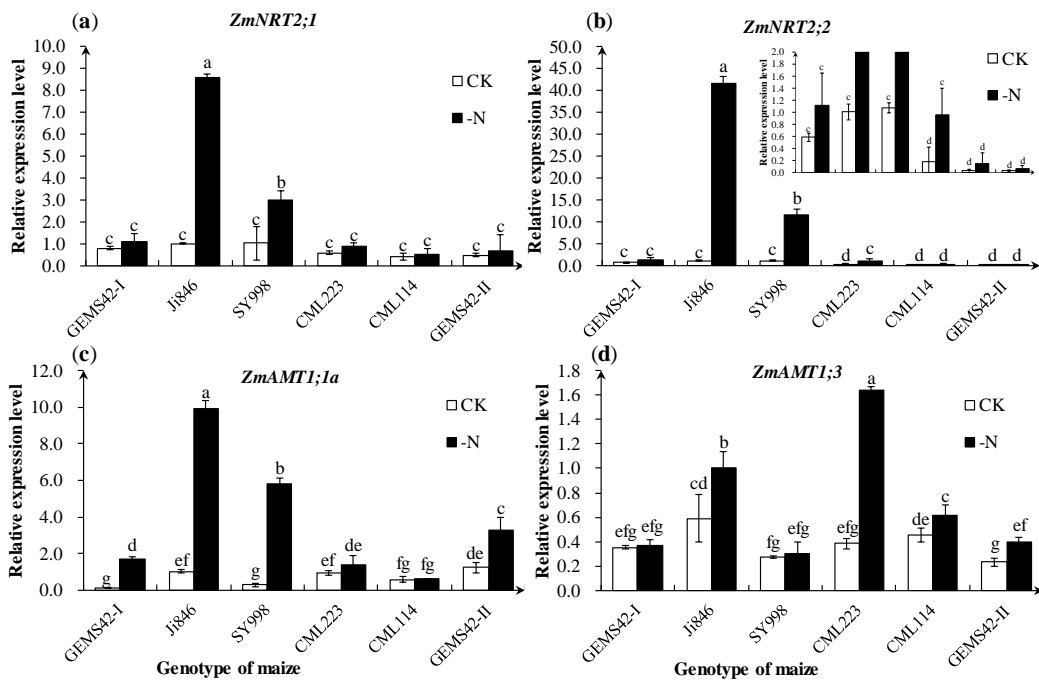

**Figure 4** **Effects of low nitrogen stress on nitrogen transporter genes of maize roots.** Effects of low nitrogen stress on nitrogen transporter genes of maize roots. (A) *ZmNRT2;1*, (B) *ZmNRT2;2*, (C) *ZmAMT1;1a*, (D) *ZmAMT1;3* . The +N and –N represent the seedling under high nitrogen (3 mM $NH_4NO_3$) and low nitrogen (0.3 mM $NH_4NO_3$), respectively. Values represent the mean ±SD of ten seedlings in each treatment. Bars with different letters show significant differences at $p < 0.05$.

**Table 2** **Eigenvalue, proportion and cumulative of the first four principal components based on the nitrogen-tolerant index of maize.**

| Index | Principal component | | | |
|---|---|---|---|---|
| | **1** | **2** | **3** | **4** |
| Eigenvalue | 8.054 | 3.676 | 2.627 | 2.091 |
| Proportion of variance explained (%) | 47.376 | 21.622 | 15.455 | 12.299 |
| Cumulative variance explained (%) | 47.376 | 68.997 | 84.453 | 96.751 |
| Weight coefficient $W(i)$ | 0.49 | 0.22 | 0.16 | 0.13 |

analysis results (Table 3). Then, the principal component value and subordinate function value were calculated. Finally, each maize has a D value (Table 3). The SY998, GEMS42-I and GEMS42-II have the higher D value that can define as low nitrogen tolerant maize, while the Ji846 with a lower D value was defined as low nitrogen sensitive maize (Table 3).

## DISCUSSION

Different maize performs quite differently in the complex physiology and development of roots and shoots in response to nitrogen limitation (*Hirel et al., 2001*; *Giehl, Gruber & von Wirén, 2014*). Investigation of the economic yield of crops under nitrogen deficient

Miao et al. (2022), PeerJ, DOI 10.7717/peerj.14218

**Table 3  The D value of each maize.**

| Maize | Factor score (FAC i) | | | | Principal component value (X) | | | | Subordinate function value U(X i) | | | | Weight coefficient W( i) | | | | D Value |
|---|---|---|---|---|---|---|---|---|---|---|---|---|---|---|---|---|---|
| | FAC1 | FAC2 | FAC3 | FAC4 | X1 | X2 | X3 | X4 | U1 | U2 | U3 | U4 | W1 | W2 | W3 | W4 | |
| GEMS42-I | 0.17 | 1.72 | −0.30 | 0.40 | 0.48 | 3.29 | −0.49 | 0.58 | 0.68 | 1.00 | 0.47 | 0.79 | | | | | 0.74 |
| Ji846 | −1.88 | −0.42 | 0.56 | 0.12 | −5.34 | −0.81 | 0.91 | 0.18 | 0.00 | 0.25 | 0.78 | 0.70 | | | | | 0.27 |
| SY998 | 0.42 | 0.50 | 1.17 | 0.18 | 1.18 | 0.96 | 1.90 | 0.26 | 0.77 | 0.57 | 1.00 | 0.72 | 0.49 | 0.22 | 0.16 | 0.13 | 0.75 |
| CML223 | 0.07 | −0.61 | −1.52 | 1.01 | 0.19 | −1.16 | −2.64 | 1.46 | 0.65 | 0.18 | 0.00 | 1.00 | | | | | 0.49 |
| CML114 | 0.12 | −0.07 | −0.63 | −1.93 | 0.33 | −0.14 | −1.02 | −2.79 | 0.67 | 0.37 | 0.36 | 0.00 | | | | | 0.47 |
| GEMS42-II | 1.11 | −1.12 | 0.72 | 0.22 | 3.16 | −2.15 | 1.16 | 0.32 | 1.00 | 0.00 | 0.84 | 0.73 | | | | | 0.72 |

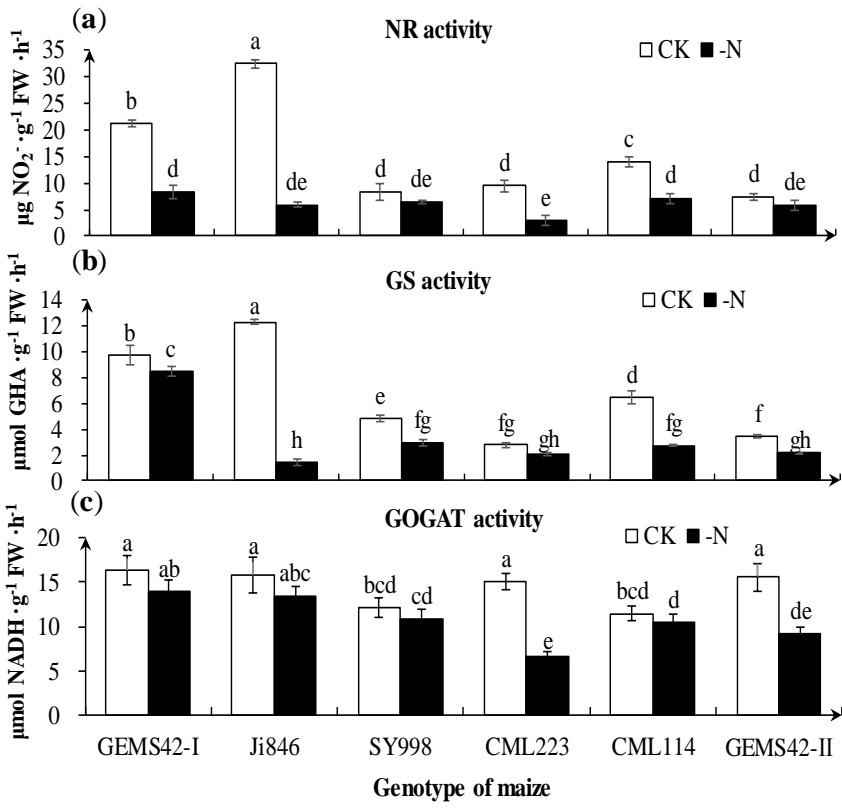

**Figure 5** **Effects of low nitrogen stress on nitrogen metabolism enzymes of maize roots.** (A) Nitrate reductase (NR) activity, (B) Glutamine synthetase (GS) activity, (C) Glutamate synthase (GOGAT) activity. The +N and –N represent the seedling under high nitrogen (3 mM $NH_4NO_3$) and low nitrogen (0.3 mM $NH_4NO_3$), respectively. Values represent the mean ± SD of ten seedlings in each treatment. Bars with different letters show significant differences at $p < 0.05$.

soil is the most and objective method for screening low nitrogen tolerant plants. However, this method that covers the whole growth period is time and labor-consuming. Therefore, the effective evaluation of indicators of different maize genotypes are issue to be explored. The morphological and physiological characteristics are widely used for low nitrogen tolerant crop screening at the seedling stage. The root morphology changes affect the nitrogen efficiency through the alteration of nitrogen absorption. The relative nitrogen uptake could be an indicator of the low nitrogen tolerance evaluation of barley (*Jiang et al., 2019*). The root architecture and function that contribute to nitrogen absorption efficiency (*Trachsel et al., 2011*), and the morphology of roots were also closely associated with the acquisition of nitrogen and the development of plant shoots (*Mi et al., 2010*; *Lynch, 2013*; *Li et al., 2017*). The root and shoot biomass of Ji846 was significantly decreased under the nitrogen limitation (Figs. 1B and 1C). In addition, the root total length, root surface and root volume were significantly decreased in Ji846 under low nitrogen conditions (Fig. 2B, 2d and 2e). Ji846 has the lowest nitrogen accumulation under nitrogen limitation (Table 1). These results indicated that Ji846 was potential nitrogen inefficient maize. The
shoot and root fresh weight of SY998, GEMS42-I and GEMS42-II have not significantly inhibited by nitrogen limitation, which exhibited high nitrogen efficiency to maintain plant growth (Fig. 1). The higher nitrogen accumulation of SY998, GEMS42-I and GEMS42-II were observed under nitrogen limitation (Table 1). Therefore, SY998, GEMS42-I and GEMS42-II may be the potential nitrogen-efficient maize. The root total length, root surface and root volume were increased in SY998 and GEMS42-II in response to nitrogen limitation (Fig. 2B, 2d and 2e). This consist with the previous study that plant shoots could be associated with nitrogen efficiency in selecting for improving grain yield under low nitrogen conditions (*Chen et al., 2016*). Recent studies show that the root weight and root length of nitrogen-tolerant maize were significantly increased in response to nitrogen limitation (*Singh et al., 2022*). In addition, the D value of the three maize is higher than 0.7, while the D value of Ji846 (potential nitrogen inefficient maize) is 0.27 in the present study (Table 3). Therefore, according to D value for low nitrogen tolerance evaluation was consistent with the physiological indicators. In addition, the complicated traits were simplified to reflect the low nitrogen tolerance information of maize by introducing the D value to evaluate stress tolerance.

NRT2 belongs to the high-affinity nitrate transporter. Previous research indicated that the transcript levels of *ZmNRT2* were induced by low nitrogen (*Santi et al., 2003*; *Liu et al., 2009*). However, another study showed that the baseline transcript levels of *ZmNRT2.1* and *ZmNRT2.2* were generally much higher than for any of the other transporters, regardless of the external (*Garnett et al., 2013*). ZmAMT1;1a and ZmAMT1;3 are most probably the major components in the high-affinity transport system in maize roots (*Gu et al., 2013*). Interestingly, the higher expression of *ZmNRT2;1*, *ZmNRT2;2*, *ZmAMT1;1a* and *ZmAMT1;3* in Ji846 not increased its nitrate and ammonium content under nitrogen limitation (Figs. 3 and 4). Ji846 even has the lowest nitrogen accumulation under nitrogen limitation (Table 1). Among the three higher D value maize, the expression of *ZmNRT2;1* and *ZmNRT2;2* only significantly increased in SY998 not in GEMS42-I and GEMS42-II (Fig. 4 and Table 3). All of the three maize have higher nitrogen accumulation under nitrogen limitation, especially SY998 has the highest accumulation (Table 1). The expression levels of *ZmNRT2;1*, *ZmNRT2;2*, *ZmAMT1;1a* and *ZmAMT1;3* were not correlative with nitrogen content in maize. On the other hand, some other uptake systems might exist in maize for nitrogen absorption. Therefore, the evaluation of nitrogen efficiency by these genes was inappropriate, at least in maize seedlings.

Nitrate ($NO_3^-$) and ammonium ($NH_4^+$) taken up by plants must first be assimilated into amino acids before it can be used for protein synthesis for plant growth. Hence the nitrogen-assimilation enzyme is a feasible strategy for improving nitrogen efficiency. NR is the first enzyme to reduce the $NO_3^-$ to $NO_2^-$, and further reduce to $NH_4^+$ by nitrite reductase (*Lea et al., 2006*; *Takahashi et al., 2001*). The $NH_4^+$ is assimilated into amino acid by the GS-GOGAT cycle, which is a crucial step for converting inorganic nitrogen into organic nitrogen in plants (*Martin et al., 2006*). The NR and GS activities of JI846 decreased by over 80%, while the NR activities decreased by 20% and GS activities decreased by 30% both in SY998 and GEMS42-II under low nitrogen conditions (Fig. 5). Therefore, the potential low nitrogen tolerant maize varieties (SY998 and GEMS42-II) have

greater enzyme activities as compared to potential nitrogen inefficient maize. The high nitrogen efficient genotypes also had more enzyme activities than low nitrogen inefficient genotypes in barley in response to nitrogen limitation (*Shah et al., 2017*). Also, the higher nitrogen utilization efficiency rice has a higher nitrogen-assimilation enzyme activity (*Yi et al., 2019*).

Therefore, according to the D value, low nitrogen tolerance evaluation was feasible. In addition, the complicated traits were simplified to reflect the low nitrogen tolerance information of maize by introducing the D value to evaluate stress tolerance. Based on the D value, SY998 and GEMS42-II were potential low nitrogen tolerant maize and nitrogen efficient genotypes, and Ji846 was potential low nitrogen sensitive maize and nitrogen inefficient genotype. Further study should be conducted to verify the yield and heritability effects of these genotypes in the field.

## CONCLUSIONS

Low nitrogen tolerance of maize is a complex trait that is determined by both genetic and environmental factors. Seventeen traits of six maize genotype related to nitrogen were investigated and a D value was introduced to screen potential low nitrogen-tolerant maize in the present study. The potential nitrogen-efficient maize (SY998, GEMS42-I and GEMS42-II) that had a higher D value (above 0.7) showed better growth performance. In contrast, the potential nitrogen inefficient maize (Ji846) had the lowest D value (0.27) with significant growth inhibition in response to nitrogen limitation. Therefore, using the D value to comprehensively evaluate low nitrogen tolerance can integrate the multiple nitrogen-related traits, which can avoid the one-sidedness of a single parameter. Since the D value was calculated based on the theory of fuzzy mathematics. This method may also provide the benefit of development techniques to screen other potential stress-tolerant traits.

### Funding

Shenyang Agricultural University Talent Introduction Scientific Research Project (Grant No. 2016026) The funders had no role in study design, data collection and analysis, decision to publish, or preparation of the manuscript.

### Grant Disclosures

The following grant information was disclosed by the authors:
Shenyang Agricultural University Talent Introduction Scientific Research:  2016026.

### Competing Interests

The authors declare there are no competing interests.

## Author Contributions

- Jianjia Miao conceived and designed the experiments, performed the experiments, analyzed the data, prepared figures and/or tables, authored or reviewed drafts of the article, and approved the final draft.
- Fei Shi performed the experiments, analyzed the data, prepared figures and/or tables, and approved the final draft.
- Wei Li performed the experiments, analyzed the data, prepared figures and/or tables, and approved the final draft.
- Ming Zhong conceived and designed the experiments, authored or reviewed drafts of the article, and approved the final draft.
- Cong Li conceived and designed the experiments, authored or reviewed drafts of the article, and approved the final draft.
- Shuisen Chen conceived and designed the experiments, analyzed the data, authored or reviewed drafts of the article, and approved the final draft.

## Data Availability

The sequences of the primers used for real-time PCR are available in Table S1. The gene sequences in Table S1 are also available at MaizeGDB.

## Supplemental Information

Supplemental information for this article can be found online at http://dx.doi.org/10.7717/peerj.14218#supplemental-information.

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
