# Peer review of "Comprehensive screening of low nitrogen tolerant maize based on multiple traits at the seedling stage"

_PeerJ, doi:10.7717/peerj.14218_

## Round 0.1 · original submission · Major Revisions

The manuscript "Comprehensive screening of low nitrogen tolerant maize based on multiple traits at the seedling stage" attempts to screen stress-tolerant of seventeen nitrogen-related traits of six maize genotypes. The D value was introduced in this study to assess stress tolerance. This method has the potential to reduce the complexity of the investigated traits while increasing the accuracy of stress tolerant evaluation. Furthermore, this method not only quickly screens potentially tolerant germplasm for low nitrogen tolerance, but also includes as many correlated traits as possible to avoid the one-sidedness of a single parameter. This manuscript's information is valuable, but there are some major issues that must be addressed before it can be published in this esteemed journal. Based on the comments, we decided that your manuscript could be reconsidered for publication if you are willing to make significant revisions.

Reviewer 2 has suggested that you cite specific references. You are welcome to add it/them if you believe they are relevant. However, you are not required to include these citations, and if you do not include them, this will not influence my decision.

·

Basic reporting

Comments to authors
The manuscript “Comprehensive screening of low nitrogen tolerant maize based on multiple traits at the seedling stage” deals with the screening of low nitrogen tolerant maize with the development of a mathematical model with a D value which represents the tolerant level of the maize cultivar. The manuscript is well planned and executed. However, some concerns need to be addressed before the final decision is made.
General comments:
• There are some typographical errors that need to be looked into.
• In the introduction section, the objective of the research needs to provided in more detail
• The materials & Methods and results sections are well written and appreciated.
• The discussion section needs to be improved
Comments
1. The authors need to describe the results in the abstract section more clearly.
2. LN 40: The author must provide the latest reference here.
3. LN 46-47: Please rewrite the line.
4. LN 86-90: Check the lines and rewrite it.
5. LN 161: What was the basis of taking “Low nitrogen (ca. 0.3 mM N)”?
6. LN 165: Fig 2b is cited first. Kindly cite all figures and tables in proper sequence
7. In result section, the authors should mention the percentage increase or decrease in value. This will give a clear picture there the treatment enhanced/reduced the particular.
8. ¬¬¬The section “Enzyme activity of the key enzymes referring to nitrogen metabolism” Is well written.
9. The discussion section needs to be written more in detail by adding facts on the aspect of calculation of D value and its correlation with the low nitrogen tolerance.
10. The future impact of the present study may be added in the conclusion section.

Experimental design

The experimental design is well designed and will be highly helpful for plant breeders and physiologists in screening cultivars based on tolerance and sensitive level of different nutrient levels.

Validity of the findings

The data presented in the current manuscript is well analyzed and described. However, some minor details are required in respect of the discussion about D values.

Reviewer 2 ·

Basic reporting

No comments

Experimental design

No comments

Validity of the findings

No comments

Additional comments

Line 17; rewrite “screening in different investigation”
Line 24 to 26 “overall rewrite results in abstract section also rewrite the results “In contrast, maize with lowest D value was cons……”
Line 37 to 39 provide suitable reference
The aims of this study rewrite end of introduction.
Why you select these 6 genotypes for this study? Any significant reason?
Line 92 to 95 provide suitable reference at the end of line https://doi.org/10.1007/s42729-021-00720-9
Discussion is very short. Please add more reports in the discussion section.
Line 140 and156 “Sample variability was expressed as the standard error of the mean” you have written standard error, but in the figures, legend you are mentioned SD (Standard deviation). Clear this confusion.
Grammatical and typo errors should be corrected
-Correct formatting of some references cited in text
-Abbreviations should be rechecked to follow the standard format (full form should not be repeated)

·

Basic reporting

no comment

Experimental design

Authors taken only six maize genotypes to screen and validate tolerant maize against low nitrogen based on multiple traits at seedling stage. For validating such a important parameter, the number of genotypes should be more (at least 100). Please justify.

Validity of the findings

1. In abstract, background lacks information regarding presented research work.
2. In result section of abstract, authors should mention the name of genotypes tolerant against low nitrogen.
3. What is D value, please mention in abstract and introduction.
4. The discussion part is shallow and needs additional recent reference to support the results.
5. Conclusion part needs rewriting as it only explained the importance of D value. The finding of result should be mentioned in this section.

Additional comments

Line 197: Write subheading as “Nitrogen metabolism related key enzymes activity assay”.

---

## Round 0.2 · Major Revisions

Although the authors addressed the reviewers' concerns thoroughly, there is still space for improvement in both the English language and the presentation style. Please see the attachment for more information.

·

Basic reporting

The suggested modification has been made in the manuscript.

Experimental design

NA

Validity of the findings

NA

---

## Round 0.3 · accepted · Accept

The authors have sincerely addressed the reviewers comments.

Reviewer 2 ·

Basic reporting

Acceptable in the present form.

Experimental design

nil

Validity of the findings

mil

Additional comments

nil

·

Basic reporting

Suggested comments has been incorporated in the revised manuscript.

Experimental design

No comment.

Validity of the findings

No comment.